# Genetic Potential of Newly Developed Maize Hybrids under Different Water-Availability Conditions in an Arid Environment

**DOI:** 10.3390/life14040453

**Published:** 2024-03-29

**Authors:** Youstina S. A. Sedhom, Hassan A. Rabie, Hassan A. Awaad, Maryam M. Alomran, Salha M. ALshamrani, Elsayed Mansour, Mohamed M. A. Ali

**Affiliations:** 1Department of Crop Science, Faculty of Agriculture, Zagazig University, Zagazig 44519, Egypt; ysedhom@zu.edu.eg (Y.S.A.S.); hrabie@zu.edu.eg (H.A.R.); hawaad@zu.edu.eg (H.A.A.); mohammed_ali@zu.edu.eg (M.M.A.A.); 2 Department of Biology, College of Science, Princess Nourah bint Abdulrahman University, Riyadh 11671, Saudi Arabia; 3Department of Biology, College of Science, University of Jeddah, Jeddah 21959, Saudi Arabia; smalshmrane@uj.edu.sa

**Keywords:** cluster analysis, drought stress, genetic diversity, heterosis, principal component analysis, tolerance indices

## Abstract

Drought is a crucial environmental stress that tremendously impacts maize production, particularly under abrupt climate changes. Consequently, breeding drought-tolerant and high-yielding maize hybrids has become decisive in sustaining its production and ensuring global food security under the global fast-growing population. The present study aimed to explore drought tolerance and agronomic performance of newly developed maize inbred lines and their hybrids. Ten newly developed maize inbred lines were crossed with two high-yielding testers using a line × tester mating design. The developed twenty hybrids alongside two high-yielding commercial hybrids were evaluated under water-deficit (5411 m^3^/ha) and well-watered (7990 m^3^/ha) conditions in dry summer climate conditions. Highly significant variations were detected among the evaluated hybrids for all studied agronomic traits under well-watered and water-deficit conditions. The inbred lines L10 and L6 were particularly notable, demonstrating the most significant negative general combining ability (GCA) effects for earliness, which is crucial for stress avoidance in both environmental settings. Inbred lines L11, L7, L6, and L1 also showed the highest positive and most significant GCA effects for key yield traits, indicating their potential as parents in breeding programs. The crosses L-10×T-1 and L-6×T-2 were outstanding for their heterotic effects on earliness in days to tasseling and silking. Similarly, the crosses L-4×T-2 and L-1×T-1 excelled in plant and ear heights under both irrigation regimes. The hybrids L-1×T-2 and L-7×T-1 demonstrated superior heterosis for chlorophyll content, number of rows per ear, and overall grain yield. Additionally, hybrids L-11×T-1 and L-11×T-2 exhibited remarkable heterotic effects for the number of grains per row, number of rows per ear, 100-kernel weight, and grain yield, highlighting their potential in breeding for productivity. Based on drought tolerance indices and cluster analysis, the cross combinations L-11×T-1, L-11×T-2, L-7×T-1, and L-1×T-2 were classified as the most drought-tolerant crosses. The principal component analysis highlighted traits such as days to tasseling, days to silking, chlorophyll content, plant height, ear height, number of grains per row, number of rows per ear, and 100-kernel weight can be taken as selection criteria for improving grain yield in maize breeding programs under limited water conditions. Based on the summarized results, the identified genetic materials could be considered promising under both conditions and hold potential for future breeding programs.

## 1. Introduction

Maize (*Zea mays*) is a main source of food, feed, and fuel production globally, ranking third after wheat and rice [1]. Its harvested area reached 203.4 million hectares in 2022 and produced 1163.5 million tons [2]. The cultivated area in Egypt was 930 thousand hectares in 2022, producing 7.6 million tons [2]. However, there is a substantial gap between the consumption and production of maize grains. Consequently, elevating the yield potential of this maize remains the primary goal of Egyptian breeders to narrow this disparity. Hence, maize breeders are attempting to develop new maize hybrids to exploit heterosis and select promising ones for commercial production [3].

Water scarcity represents the main constraint facing crop productivity in arid regions [4]. Drought stress adversely impacts the anthesis-silking interval, grain yield, and yield-contributing traits in maize [5]. The harmful impacts on agronomic traits due to water stress become more evident when the plants are exposed to water deficit during the flowering stage, which exacerbates yield reduction [6]. Breeding programs could be effectively used in developing new maize hybrids with high-yield productivity under water scarcity [7]. Over the years, significant progress has been achieved in elevating maize yields under drought stress. However, progress has been slow because drought is a complex quantitative trait governed by multiple genes and pathways; it is strongly affected by genotype−environment effects and low heritability [8]. Hence, developing superior maize genotypes with a high degree of tolerance to water stress becomes pivotal to breeders for sustainability.

Heterosis remains a crucial parameter in plant breeding and could be employed to develop varieties with higher yields, improved quality, and tolerance to environmental stress [9]. Through hybrid vigor, plant adaptability to adverse environmental conditions, such as drought and temperature extremes, can also be increased [10]. Identified superior hybrids can be further evaluated for commercial viability, potentially augmenting farmers’ incomes and mitigating poverty, hunger, and malnutrition in Africa [11]. The heterosis used by breeders is essential in categorizing maize lines into specific heterotic groups. Exploiting the heterosis phenomenon in maize is essential in developing new promising maize hybrids with increased yield potential and surpassing prevailing commercial hybrids in agronomic traits [12]. Moreover, capitalizing on hybrid vigor and developing new hybrids are imperative in sustainable agriculture by mitigating the adverse effects of environmental stresses and boosting food production efficiency to meet escalating societal demands [13,14]. The line × tester mating scheme is an effective method for estimating both general (GCA) and specific (SCA) combining ability effects, as well as for identifying suitable parents [15]. This approach is efficient in selecting superior parents for developing high-yielding hybrids and tolerant to environmental stresses [16]. Therefore, the current work was undertaken to explore the drought tolerance and agronomic performance of ten newly developed maize inbred lines and their twenty hybrids, to identify superior drought-tolerant hybrids exhibiting enhanced heterosis under limited water conditions, and explore the association among studied agronomic traits under water-deficit conditions.

## 2. Materials and Methods

### 2.1. Parental Genotypes and Hybridization

The present study employed inbred lines that were newly developed from various genetic resources in Egypt alongside introduced lines from the International Maize and Wheat Improvement Center (CIMMYT) (Table 1). Ten inbred lines and two testers were selected based on their diversity from earlier screening trials. The parental lines are tropical genotypes of the dent corn characterized by their white kernels. In the summer of 2019, the selected ten lines and the two testers were cultivated on three sowing dates to account for differences in flowering times. At the flowering stage, the ten inbred lines were hybridized with the two testers utilizing the line × tester mating design, providing twenty top crosses.

### 2.2. Experimental Design and Agronomic Practices

The twenty developed top crosses alongside their parental lines (ten lines and two testers) and two high-yielding check hybrids (SC-10 and SC-30k8) have been adapted and widely cultivated across various Egyptian environments. The studied genotypes were evaluated under two irrigation regimes during the summer season of 2020 at the Experimental Farm of Benha University, Egypt (30°21′ N, 13°26′ E). The experimental site represents Egypt’s characteristic climate, predominantly hot and arid (Appendix A). Throughout the maize-growing season, the country experiences an absence of precipitation events, underscoring the challenging environmental conditions for agriculture. The soil at the experimental site is clay textured (28.67% sand, 20.10% silt, and 51.23% clay) with middle alkaline properties (pH = 7.9); the properties are detailed in Appendix A. The developed hybrids and their parental lines underwent assessment under two irrigation treatments: well-watered and water-defict conditions. Furrow irrigation was applied following the practice of the region of the study.

The Department of Water Requirement and Field Irrigation, Center of Agricultural Research under the Egyptian Ministry of Agriculture and Land Reclamation, determines study region recommended irrigation practices for maize. The optimal irrigation amount was identified at approximately 7900 m^3^/ha based on climatic variables and soil type. Accordingly, irrigation was applied to well-watered treatment every 12 days, providing a total of 7990 m^3^/ha throughout the growing season. Conversely, irrigation in stressed treatment was applied every 21 days, supplying 5411 m^3^/ha, which represents about 65% of the recommended amount to induce water-deficit conditions. The applied irrigation amount at each regime was measured using a flow meter. A 6-m alley separated the irrigation regimes to control water movement. The strip-plot experiment in a randomized complete block design with three replications was applied to the field experiment. The horizontal plots assigned irrigation treatments, while the vertical plots had maize genotypes and were distributed randomly. Each plot consisted of three ridges 3 m long with a 0.70-m space between the ridges. These distances provided an experimental plot size of 6.3 m^2^. Each hill was spaced 0.25 m apart, with one plant per hill. The sowing date was the first of May, which is the region’s optimal maize cultivation period. Three seeds were sown per hill, and the strongest seedling was retained at full emergence (20 days after sowing). Prior to sowing, phosphorus and potassium fertilizers were applied in a single dose at rates of 115 kg K_2_O/ha (48% K_2_O) and 80 kg P_2_O_5_/ha (15.5% P_2_O_5_), respectively. Nitrogen fertilization was applied at 276 kg of nitrogen per hectare in two equal doses at the first and third irrigation under both irrigation regimes.

### 2.3. Measured Traits

Data were recorded on days to tasseling, days to silking, plant height (cm), ear height (cm), chlorophyll content (SPAD value), number of rows per ear, number of grains per row, 100-kernel weight (g), and grain yield per hectare (ton). Data on days to 50% tasseling and days to 50% silking were recorded on plot bases. Data on ear height, plant height, chlorophyll content, number of rows per ear, number of grains per row, and 100-kernel weight were recorded on ten plants and were selected randomly from each experimental plot. Grain yield (t/ha) was estimated based on the harvested plot and converted to t/ha.

### 2.4. Statistical Analysis

The data underwent analysis of variance following the line × tester model proposed by Kempthorne [17]. Differences among maize hybrids were separated by the least significant difference test at *p* ≤ 0.05. The general combining ability (GCA) effects of the lines and testers, as well as the specific combining ability (SCA) effects of the hybrids, were computed using line × tester analysis following the methodology of Kempthorne [17]. Standard heterosis was calculated for traits showing significant differences among hybrids. It was calculated for each top cross as the percent deviation of F1 performance from either SC-10 or SC-30k8 values for each experiment. The standard heterosis (superiority) = [(F1 − Check variety)/(Check variety) × 100]. The significance of heterosis was tested using a t-test against the critical difference (CD). Appropriate CD values were computed according to the following formulae for heterosis relative to check variety = tp% × √(2MSe/r) where tp% is the tabulated t value at a stated level of probability for the experimental error degree of freedom and r is the replications number. Seven drought indices were measured for grain yield. The calculated indices were mean productivity (MP), stress tolerance index (STI), tolerance index (TOL), stress sensitivity index (SSI), yield index (YI), yield stability index (YSI), and harmonic mean (HM), as shown in Appendix A. Principal component analysis (PCA) was performed on the averages of the measured traits to explore their relationships under water-deficit conditions. Cluster and principal component analyses were performed using R statistical software version 3.6.1.

## 3. Results

### 3.1. Analysis of Variance and Mean Performance

The genetic variation of assessed plant materials was assessed through analysis of variance (ANOVA) under both irrigation treatments: well-watered and stressed conditions. Mean squares due to crosses and their partitions, lines, testers, and line × tester were significant for all studied traits under well-watered and stressed conditions (Table 2). This indicates a considerable degree of genetic diversity among the evaluated plant materials under both conditions. General combining ability (GCA) exhibited a significantly higher magnitude than specific combining ability (SCA) for all traits, indicating that these traits are primarily governed by nonadditive gene action under both irrigation treatments. Among all evaluated crosses, the highest grain yield was produced by the cross L-1×T-2, L-11×T-2, L-7×T-1, and L-11×T-1 under well-watered and water-stress conditions. Also, the abovementioned top crosses were the best among all studied hybrids for the number of rows per ear, number of grains per row, and 100 kernel weight under well-watered and stressed conditions (Appendix A). The cross L-4×T-2 displayed the shortest plant height, recording 195.3 and 193.7 cm under respective conditions compared to the other hybrids (Appendix A). Additionally, the cross L-1×T-1 demonstrated the most desirable ear height, registering 88.0 and 86.7 cm under well-watered and stressed conditions, respectively (Appendix A). Likewise, the cross combination L-10×T-1 exhibited favorable early tasseling and silking among all evaluated crosses (Appendix A). It expressed the best values of 50.7 and 51.7 for days to tasseling and 58.3 and 60.7 for days to silking under well-watered and water-deficit conditions, respectively. Superior chlorophyll content was recorded by the cross L-1×T-2 under well-watered treatment (61.7) compared to the other hybrids. The cross L-11×T-2 expressed the best mean value of this trait under water stress. Accordingly, the three single top crosses, L-7×T-1, L-1×T-2, and L-11×T-2, expressed the best values for grain yield, number of rows per ear, number of grains per row, 100-kernel weight, and chlorophyll content, under both conditions. This revealed a high degree of dissimilarity between T-1 and L-7, and between T-2 and each of L-1 and L-11. Consequently, these three crosses are prospective and can be utilized in future breeding programs.

### 3.2. Combining Ability

General combining ability effects for all traits under well-watered and stressed conditions are presented in Figure 1 and Figure 2. The parental inbred lines L11, L7, L6, and L1 exhibited the highest positive and significant GCA effects for grain yield. Line 11 demonstrated superiority in the number of rows per ear under both conditions (Figure 1). Lines L-11, L-10, and L-9 expressed the most desirable GCA effects for the number of grains per row under both irrigation treatments. L11, L7, L1, and T2 recorded the highest and most significant GCA effects for 100-kernel weight. The lines L1, L2, and L4 as well as tester T1, exhibited desirable GCA effects for plant and ear heights under both irrigation treatments. Notably, the inbred lines L10 and L6 expressed significant negative GCA effects values for days to tasseling and days to silking under well-watered and stressed conditions (Figure 2). Specific tester T1 exhibited the most desirable GCA effects for days to silking. Lines L1, L6, and L11 had the highest GCA effects for chlorophyll content under both treatments.

Table 3 presents the specific combining ability (SCA) effects for the studied traits under well-watered and stressed conditions. The crosses L-5×T-1, L-7×T-1, L-1×T-2, and L-10×T-2 recorded the highest positive SCA effects for grain yield under both conditions. Moreover, L-7×T-1, L-1×T-2, L-5×T-2, and L-8×T-2 recorded the highest positive SCA effects for the number of rows per ear under both environments. Additionally, L-4×T-1, L-1×T-2, L-8×T-2, L-9×T-2, L-10×T-2, and L-11×T-2 exhibited the highest positive SCA effects for number of grains per row. Furthermore, L-7×T-1, L-8×T-1, L-9×T-1, L-10×T-1, L-11×T-1, L-1×T-2, and L-6×T-2 showed the highest positive SCA effects for 100-kernel weight. Additionally, L-1×T-1, L-10×T-1, L-4×T-2, and L-5×T-2 showed desirable negative SCA effects for plant and ear heights. Notably, the top crosses L2×T1, L-1×T-2, and L-9×T-1 exhibited the most desirable negative SCA effects for days to tasseling and days to silking under both conditions. Furthermore, the crosses L-7×T-1, L-1×T-2, L-6×T-2, and L-9×T-2 demonstrated the highest positive SCA effects for chlorophyll content.

### 3.3. Standard Heterosis

The crosses L-7×T-1, L-11×T-1, L-1×T-2, and L-11×T-2 exhibited desirable heterosis for grain yield under both environments relative to SC-10 and SC-30k8 (Table 4). The cross L-1×T-2 recorded the highest heterosis for this trait relative to SC-10 and SC-30k8 under both environments. The cross L-7×T-1 exhibited the highest heterosis for grain yield relative to SC-10 and SC-30k8 under both irrigation treatments, followed by L-11×T-2 (Table 4). Desirable positive and significant heterosis was detected for the number of rows per ear by the cross combinations L-1×T-2, L-11×T-2, and L-4×T-2 under well-watered conditions, while L-11×T-1, L-4×T-2, and L-11×T-2 under stressed conditions relative to SC-10 and SC-30k8 (Table 4). The best heterosis for the number of rows per ear was obtained for the cross L-1×T-2 relative to SC-10 and SC-30k8 under both environments. The positive and significant heterosis for the number of grains per ear was obtained by the cross combinations L-11×T-2 followed by L-11×T-1 and L-10×T-2 relative to SC-10 and SC-30k8 under well-watered and stressed conditions (Table 4). The values of standard heterosis of 100-kernel weight showed that three cross combinations L-7×T-1, L-11×T-1, and L-11×T-2 expressed positive heterosis values relative to SC-10 and SC-30k8 under both conditions (Table 4). Five and two hybrids had desirable heterosis for plant height relative to SC-10 and SC-30k8 under both environments (Table 5). However, the best heterosis for plant height was registered for the cross L-4×T-2 relative to SC-10 and SC-30k8 under well-watered and stressed conditions. Eleven and seven crosses showed negative and significant desirable heterosis for ear height compared to SC-10 under well-watered and stressed conditions, respectively. However, the best heterosis was obtained by L-1×T-1 relative to SC-10 and SC-30k8 (Table 5). Five and six cross combinations expressed significant negative heterosis for days to tasseling relative to SC-10 under well-watered and stressed conditions, respectively (Table 5). Furthermore, relative to SC-30k8, one and two crosses exhibited significant heterotic effects for days to tasseling. The most favorable heterotic effects for days to tasseling were detected for the cross L-10×T-1 relative to SC-10 and SC-30k8 commercial hybrids. The desired negative and significant heterotic effects for days to silking were recorded by nine and seven crosses relative to SC-10 (Table 5). Two and one crosses were identified relative to SC-30k8 under well-watered and stressed conditions, respectively. Generally, the best heterosis for days to silking was detected for the cross L-7×T-1 under well-watered conditions, and L-10×T-1 exhibited superior performance under stressed conditions relative to both commercial hybrids SC-10 and SC-30k8.

Desirable positive and significant heterosis for chlorophyll content was detected by the cross combinations L-7×T-1, L-1×T-2, and L-6×T-2 under well-watered conditions, and for L-7×T-1, L-1×T-2, L-6×T-2, and L-11×T-2 under stressed conditions (Table 6). Also, desirable heterosis relative to SC-30k8 was obtained for the crosses L-7×T-1 and L-1×T-2 under well-watered conditions, and for L-1×T-2, L-6 ×T-2, and L-11×T-2 under stressed conditions. However, the best heterotic effects for chlorophyll content were recorded by the cross L-1×T-2 relative to SC-10 and SC-30k8 under well-watered and stressed conditions.

### 3.4. Genotypic Classification Based on Drought Tolerance

Based on tolerance indices (Table 7), the evaluated maize crosses were classified into three groups with subclusters using hierarchical clustering (Figure 3). Group A comprised four crosses with the highest observed grain yield and tolerance indices: L-11×T-1, L-11×T-2, L-7×T-1, and L-1×T-2. Therefore, they are considered highly tolerant crosses. Group B had nine tolerant crosses; L-5×T-1, L-6×T-1, SC-10, SC-30K8, L-8×T-1, L-2×T-1, L-6×T-2, L-9×T-1, and L-7×T-2. Group C is composed of nine crosses that had low values of tolerance indices and grain yield: L-1×T-1, L-4×T-2, L-8×T-2, L-2×T-2, L-9×T-2, L-5×T-2, L-10×T-2, L-4×T-1, and L-10×T-1 and they could be considered as sensitive crosses.

### 3.5. Association among Assessed Hybrids and Studied Traits under Drought Stress

The principal component analysis revealed that the first two principal components (PCAs) explained 65.53% of the total variation under water-deficit conditions (Figure 4). PCA1 divided the assessed hybrids based on their agronomic performance and drought tolerance. The high-yielding and tolerant hybrids were located on the positive side (L-11×T-1, L-11×T-2, L-7×T-1, and L-1×T-2). Conversely, low-yielding and sensitive hybrids (L-1×T-1, L-4×T-2, L-10×T-1, L-5×T-2, L-2×T-2, L-8×T-2, and L-10×T-2) were situated at the extreme negative end of PCA1, while moderately tolerant hybrids were positioned in the middle. Tolerant hybrids exhibited positive associations with agronomic traits, whereas agronomic traits showed negative associations with sensitive hybrids. Under stressed conditions, days to tasseling and silking showed negative associations with all yield attributes. However, a positive and significant association was observed between grain yield and all yield-contributing traits under drought conditions. Tolerance indices, harmonic mean, mean productivity, yield index, and stress tolerance index also displayed positive associations with grain yield and its components under water-deficit conditions.

## 4. Discussion

Breeding maize hybrids that are both drought-tolerant and high-yielding has become crucial for sustaining agricultural productivity in the face of shifting climate patterns and rising demands from a growing global population. The results of this study displayed significant variations among the evaluated maize lines, testers, and their crosses across well-watered and drought-stress conditions in all traits studied. This highlights a rich genetic diversity within these plant materials, offering valuable opportunities for enhancing maize resilience and water efficiency in cultivation. Crosses displayed promising agronomic performance under both well-watered and drought conditions. Remarkably, the crosses L-1×T-2, L-11×T-2, L-7×T-1, and L-11×T-1 exhibited superior grain yield, number of rows per ear number of grains per row, and 100-kernel weight under well-watered and water-stress conditions. These crosses represent promising candidates for developing drought-resilient maize hybrids. Moreover, the cross L-10×T-1 stood out for its early tasseling and silking times, while L-4×T-2, L-1×T-1, and L-1×T-2 excelled in plant height, ear height, and chlorophyll content, respectively. The selection of maize crosses tailored to varying irrigation conditions emerges as a strategic approach to bolster climate change adaptation. Likewise, Jiang et al. [5], Sah et al. [6], Badr et al. [7], Shi et al. [10], Sedhom et al. [18], and Stepanovic et al. [19] reported significant genetic variability among maize hybrids under water-limited environments. Such insights are instrumental in guiding future breeding programs to secure food production in an era of environmental uncertainty.

Mitigating the adverse effects of drought on maize yields is feasible by developing drought-tolerant hybrids. These hybrids offer the potential for large-scale cultivation in areas prone to water stress. Thereby ensuring more consistent yields and substantially mitigating the risk of yield losses due to drought conditions [20]. To identify such drought-tolerant genotypes, drought tolerance indices alongside cluster analysis serve as valuable tools, enabling the assessment of productivity under unfavorable conditions. In the present study, the maize crosses were categorized into four distinct groups with further subclusters based on hierarchical clustering that utilized tolerance indices. Notably, the crosses L-11×T-1, L-11×T-2, L-7×T-1, and L-1×T-2 emerged as highly tolerant. Consequently, employing tolerance indices and cluster analysis proves to be an effective strategy for identifying the genetic configurations of maize that are either tolerant or sensitive to environmental stresses, facilitating targeted improvements in maize breeding programs. This approach aligns with the methodologies adopted in other studies by Sedhom et al. [18], Kapoor et al. [21], Sinana et al. [22], Evamoni et al. [23], and Wang and Peng [24], who also leveraged drought indices and cluster analysis to pinpoint superior genotypes exhibiting a high degree of drought tolerance.

The results documented significant negative heterotic effects for traits such as days to silking, days to tasseling, ear height, and plant height across two different environments. The reduction in the time to silking and tasseling, along with decreased ear and plant height, is highly valued by maize breeders. These traits, collectively referred to as earliness, enable maize plants to circumvent stress from adverse environmental conditions and evade damage from pests like *Sesamia cretica*, *Pyrausta nubilialis*, and *Chilo simplex*. This finding is in harmony with the work of Panda et al. [25], Ruswandi et al. [26], and Amiruzzaman et al. [27], who observed desirable negative heterosis for earliness alongside reductions in plant and ear height. Conversely, the results identified significantly positive heterotic effects in several crosses under both well-watered and drought-stressed conditions for key productivity traits such as chlorophyll content, number of rows per ear, number of grains per ear, and 100-kernel weight. Notably, crosses L-7×T-1, L-11×T-1, L-1×T-2, and L-11×T-2 displayed pronounced positive heterosis, surpassing both commercial checks (SC-10 and SC-30k8) in performance under water-limited conditions. This suggests that these particular crosses exhibit superior drought resilience, offering the potential for enhancing maize productivity in environments challenged by water scarcity. The capability of these hybrids to perform robustly under reduced water availability underscores their potential contribution to water conservation strategies in maize cultivation. The findings align with the results of Sedhom et al. [18], Ruswandi et al. [26], Lekha et al. [28], Kahriman et al. [29], Issa et al. [30], and Adewale et al. [31], who also reported positive and significant heterotic effects for traits directly contributing to yield under stress conditions, such as chlorophyll content, number of grains per ear, number of rows per ear, 100-kernel weight, and overall grain yield. These traits are critical for developing maize hybrids capable of sustaining high productivity levels in the face of environmental challenges, thereby contributing to food security and efficient water use in agriculture.

Principal component analysis (PCA) shed light on the intricate relationships among agronomic traits under drought stress, highlighting a negative correlation between the earliness traits (days to tasseling and silking) and all measured yield attributes. This relationship suggests that selecting for earliness could concurrently benefit yield traits in environments with limited water availability. Moreover, a positive and substantial link was discovered between grain yield and all yield-related traits under drought conditions. This finding underscores the value of these traits as effective markers in maize breeding programs aimed at boosting grain yield. In addition, specific tolerance indices, including mean productivity, harmonic mean, stress tolerance index, and yield index, were found to have positive correlations with both grain yield and its components under water-scarce conditions. This indicates the practicality of these indices in pinpointing genotypes with promising drought tolerance for maize breeding efforts. The utility of these tolerance indices is further supported by the work of researchers such as Soto-Cerda et al. [32], Kumar et al. [33], Khalid et al. [34], and Amegbor et al. [35], who also employed PCA to examine the relationships between various agronomic traits.

## 5. Conclusions

Significant variations were detected in studied agronomic traits among the evaluated maize crosses, lines, and testers under both well-watered and drought-stressed conditions, highlighting the rich genetic diversity within the plant material. This diversity is a key asset for improving maize resilience and productivity in regions prone to water scarcity. Notably, the inbred lines L10 and L6 were distinguished by demonstrating the most significant and desirable general combining ability (GCA) effects for earliness, an important trait for avoiding environmental stress, under both conditions. Furthermore, inbred lines L11, L7, L6, and L1 stood out for their significant positive GCA effects on yield-related traits, underscoring their potential as foundational genetic resources for developing high-yielding maize varieties. The cross combinations L-11×T-1, L-11×T-2, L-7×T-1, and L-1×T-2 were particularly promising, showing the potential to boost yield productivity under challenging conditions while contributing to water conservation efforts. Principal component analysis (PCA) was instrumental in identifying chlorophyll content, plant and ear height, number of rows per ear, number of grains per row, and 100-kernel weight as important selection criteria for improving grain yield within maize breeding programs targeting drought-stressed environments. Moreover, the PCA biplot effectively highlighted the relevance of mean productivity, harmonic mean, stress tolerance index, and yield index for selecting genotypes with enhanced drought tolerance.

## Figures and Tables

**Figure 1 life-14-00453-f001:**
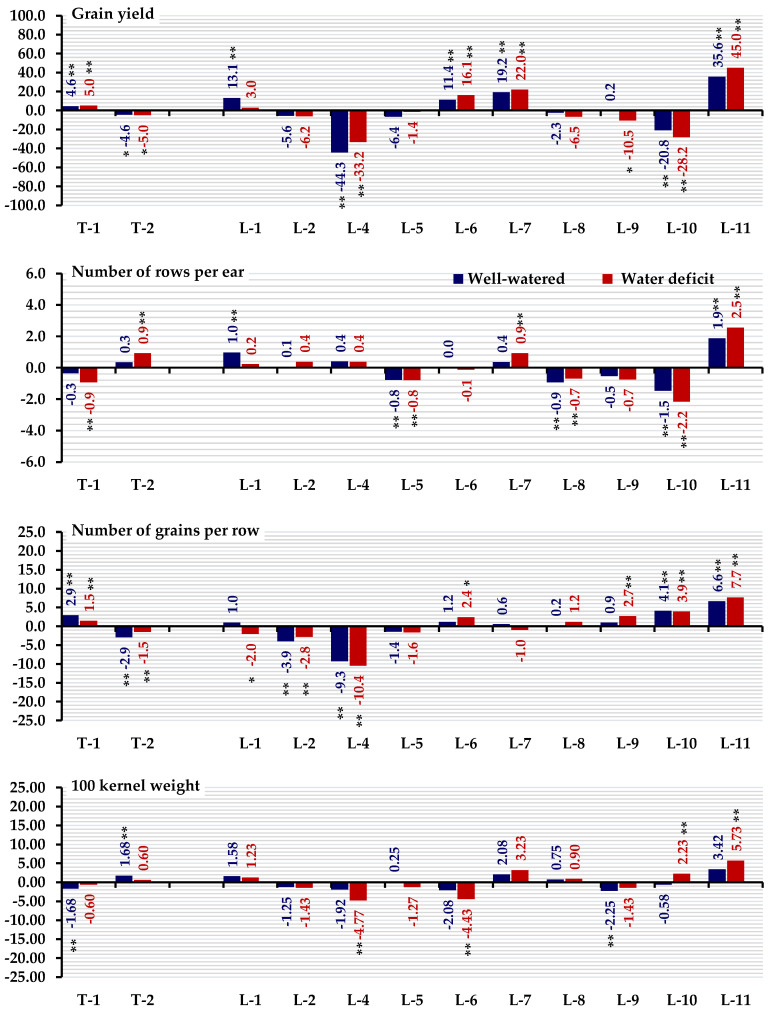
General combining ability effects (GCA) for the evaluated testers and inbred lines for grain yield, number of rows/ear, number of grains/row, and 100-kernel weight, * and ** denote significance at the 0.05 and 0.01 probability levels, respectively.

**Figure 2 life-14-00453-f002:**
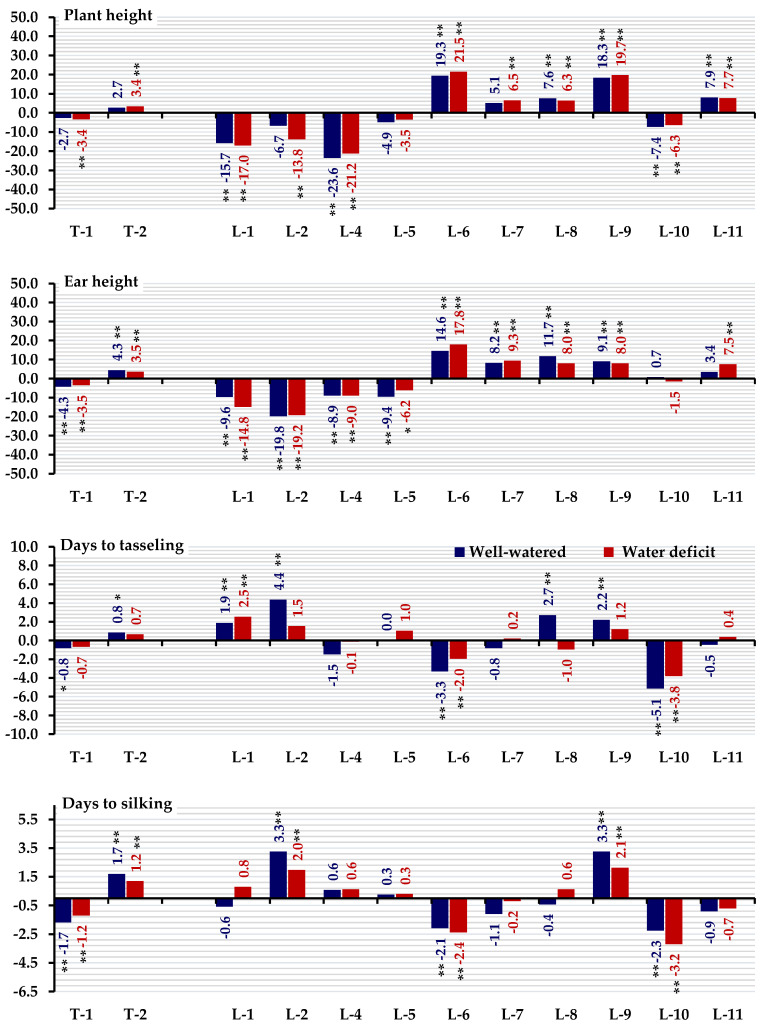
General combining ability effects for the evaluated testers and inbred lines for plant height, ear height, days to tasseling, days to silking, and chlorophyll content. * and ** denote significance at the 0.05 and 0.01 probability levels, respectively.

**Figure 3 life-14-00453-f003:**
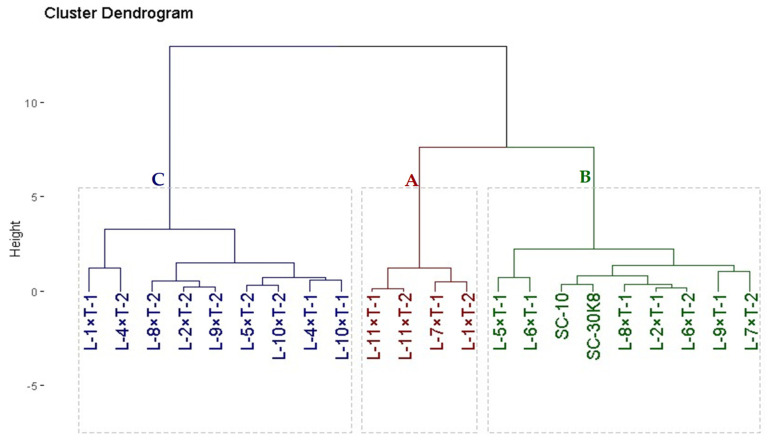
Dendrogram of the distances among developed twenty hybrids and two high-yielding commercial hybrids based on drought-tolerant indices. The hybrids were classified into three groups: A is highly drought-tolerant (four hybrids), B is moderately drought-tolerant (nine hybrids), and C is drought-sensitive (nine hybrids).

**Figure 4 life-14-00453-f004:**
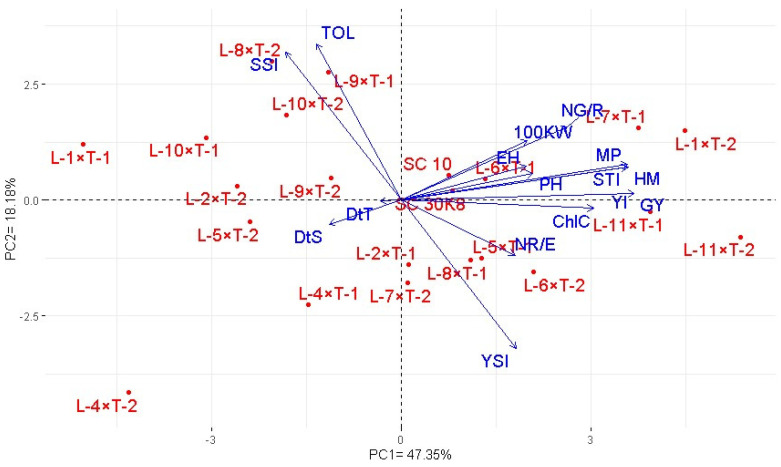
Biplot of principal component for assessed maize hybrids under drought stress based on evaluated agronomic traits under drought stress conditions and calculated tolerance indices. The high-yielding and tolerant hybrids were located on the positive side of PCA1, while low-yielding and sensitive hybrids were situated at the extreme negative end of PCA1, and moderately tolerant hybrids were positioned in the middle. Agronomic traits are associated with tolerant hybrids on the positive side of PCA1, whereas they showed negative associations with sensitive hybrids which are located on negative side of PCA1. GY: grain yield, 100 KW: 100-kernel weight, NR/E: number of rows per ear, PH: plant height, NG/R: number of grains per row, EH: ear height, DtS: days to silking, ChlC: chlorophyll content, DtT: days to tasseling, MP: mean productivity, HM: harmonic mean, TOL: tolerance index, STI: stress tolerance index, SSI: stress sensitivity index, YI: yield index, GMP: geometric mean productivity, YSI: yield stability index, and YRR: yield reduction ratio.

**Table 1 life-14-00453-t001:** Name, pedigree, origin, and some agronomic character of the ten parental inbred lines and two testers.

Code	Name	Pedigree	Origin	Kernel Type	Kernel Color	Days to Silking	Grain Yield /Plant (g)
Lines							
L-1	M5	Line developed from Cairo-1	Egypt	Dent corn	White	66.00	63.00
L-2	M8	Line developed from Giza-2	Egypt	Dent corn	White	64.00	68.17
L-4	CIMMYT-46	Line developed from La-Posta (dent population)	Mexico	Dent corn	White	64.83	64.50
L-5	CLM-19	Line developed from La-Posta (dent population)	Mexico	Dent corn	White	66.50	49.50
L-6	M26	Line developed from Giza-2	Egypt	Dent corn	White	68.33	57.67
L-7	M29	Line developed from Cairo-1	Egypt	Dent corn	White	67.83	75.67
L-8	M36	Line developed from Giza-2	Egypt	Dent corn	White	70.17	50.67
L-9	M41	Line developed from Giza-2	Egypt	Dent corn	White	69.00	64.17
L-10	M42	Line developed from Pioneer-514	Egypt	Dent corn	White	67.83	64.83
L-11	M47	Line developed from Pioneer-Fatah	Egypt	Dent corn	White	66.50	57.17
Testers							
T-1	M6	Line developed from Giza-2	Egypt	Dent corn	White	67.50	47.17
T-2	M14	Line developed from CIMMYT-14	Mexico	Dent corn	White	68.00	59.83

**Table 2 life-14-00453-t002:** Analysis of variance for studied agronomic traits of maize genotypes under well-watered and water-deficit conditions.

Source of Variance	df	Grain Yield (ton/ha)		Number of Rows/Ear		Number of Grains/Row	
		W-Watered	D-Stress	W-Watered	D-Stress	W-Watered	D-Stress
Crosses	19	8.15 **	8.96 **	4.37 **	8.71 **	134.4 **	138.30 **
Lines	9	9.58 **	10.58 **	5.74 **	9.32 **	112.2 **	141.91 **
Testers	1	4.11 **	4.96 **	7.21 **	51.34 **	510.4 **	126.44 **
Line × tester	9	7.17 **	7.77 **	2.68 **	3.37 **	114.8 **	136.01 **
Error	38	0.26	0.39	1.22	0.53	5.25	6.29
GCA		0.022	0.03	0.04	0.12	0.44	0.05
SCA		2.30	2.46	0.49	0.95	36.50	43.24
		**100-kernel weight (g)**		**Plant height (cm)**		**Ear height (cm)**	
Crosses	19	37.36 **	40.37 **	1331.2 **	1312.1 **	770.2 **	750.7 **
Lines	9	22.75 **	65.82 **	1145.4 **	1200.5 **	787.2 **	865.4 **
Testers	1	170 **	21.6 **	1092.3 **	1126.7 **	1215 **	735.0 **
Line × tester	9	37.24 **	17.01 **	1543.5 **	1444.4 **	703.7 **	637.8 **
Error	38	7.75	4.80	90.28	53.71	47.14	46.81
GCA		0.003	0.53	4.80	2.99	1.50	2.55
SCA		9.83	4.07	484.4	463.6	218.9	197
		**Days to tasseling**		**Days to silking**		**Chlorophyll content**	
Crosses	19	15.26 **	31.01 **	17.70 **	25.79 **	67.28 **	86.53 **
Lines	9	20.47 **	49.96 **	17.55 **	22.31 **	72.85 **	104.04 **
Testers	1	26.67 **	41.67 **	86.40 **	170.02 **	36.97 *	187.97 **
Line × tester	9	8.78 **	10.89 **	10.21 **	13.24 **	65.07 **	57.75 **
Error	38	5.64	3.58	2.45	2.14	8.90	6.94
GCA		0.15	0.46	0.17	0.28	0.05	0.65
SCA		1.05	2.44	2.59	3.70	18.72	16.93

* and ** denote significance at the 0.05 and 0.01 probability levels, respectively.

**Table 3 life-14-00453-t003:** Specific combining ability effects of assessed crosses for measured agronomic traits under well-watered and stressed conditions.

Hybrid	Grain Yield (t/ha)		Number of Rows/Ear		Number of Grains/Row		100 Kernel Weight (g)		Plant Height	
	W- Watered	D- Stress	W- Watered	D- Stress	W- Watered	D- Stress	W- Watered	D- Stress	W- Watered	D- Stress
L-1×T-1	−2.70 **	−2.86 **	−0.72	−1.84 **	−5.35 **	−10.32 **	−3.65 *	−1.15	−23.73 **	−19.07 **
L-2×T-1	0.02	0.52	0.05	−0.15	4.10 **	4.08 **	−1.15	0.85	−1.23	−6.90
L-4×T-1	0.39	0.40	0.05	−0.82	8.75 **	6.75 **	−2.48	0.52	33.43 **	32.93 **
L-5×T-1	0.66 *	1.02 **	−1.19	−0.99	1.67	−0.08	−0.32	1.02	16.43 **	17.60 **
L-6×T-1	0.46	0.06	0.35	−0.65	−1.38	−1.72	−2.98	−0.82	−12.90 *	−12.40 *
L-7×T-1	1.46 **	0.99 **	1.01	1.13	3.28 **	0.62	3.52 *	4.85 **	2.27	1.77
L-8×T-1	−0.01	0.84 **	−0.42	−0.75	−1.18	−2.18	2.18	1.85	5.60	6.10
L-9×T-1	0.45	0.06	0.18	−0.77	−2.62 *	−2.70 *	2.18	1.18	−10.73	−10.90 *
L-10×T-1	−0.54	−0.45	0.85	−0.62	−3.68 **	−4.62 **	1.85	2.18	−5.73	−4.57
L-11×T-1	−0.20	−0.33	−0.15	−0.32	−3.58 **	−4.48 **	0.85	0.35	−3.40	−5.23
L-1×T-2	2.70 **	2.86 **	0.72	1.84 **	5.35 **	10.32 **	3.65 *	1.15	23.73 **	19.07 **
L-2×T-2	−0.02	−0.52	−0.05	0.15	−4.10 **	−4.08 **	1.15	−0.85	1.23	6.90
L-4×T-2	−0.39	−0.40	−0.05	0.82	−8.75 **	−6.75 **	2.48	−0.52	−33.43 **	−32.93 **
L-5×T-2	−0.66 *	−1.02 **	1.19	0.99	−1.67	0.08	0.32	−1.02	−16.43 **	−17.60 **
L-6×T-2	−0.46	−0.06	−0.35	0.65	1.38	1.72	2.98	0.82	12.90 *	12.40 *
L-7×T-2	−1.46 **	−0.99 **	−1.01	−1.13	−3.28 *	−0.62	−3.52 *	−4.85 **	−2.27	−1.77
L-8×T-2	0.01	−0.84 **	0.42	0.75	1.18	2.18	−2.18	−1.85	−5.60	−6.10
L-9×T-2	−0.45	−0.06	−0.18	0.77	2.62 *	2.70 *	−2.18	−1.18	10.73	10.90 *
L-10×T-2	0.54	0.45	−0.85	0.62	3.68 **	4.62 **	−1.85	−2.18	5.73	4.57
L-11×T-2	0.20	0.33	0.15	0.32	3.58 **	4.48 **	−0.85	−0.35	3.40	5.23
**Hybrid**	**Ear height**		**Days to** **tasseling**		**Days to** **silking**		**Chlorophyll** **content**			
	**W-** **Watered**	**D-** **Stress**	**W-** **Watered**	**D-** **Stress**	**W-** **Watered**	**D-** **Stress**	**W-** **Watered**	**D-** **Stress**		
L-1×T-1	−22.83 **	−13.83 **	2.67 *	1.33	1.69 *	1.67 *	−5.18 **	−6.93 **		
L-2×T-1	4.67	8.50 *	−1.00	−1.83	−1.48	−2.48 **	−1.88	−1.03		
L-4×T-1	17.17 **	20.00 **	0.33	−0.33	−0.15	−0.82	2.70	0.04		
L-5×T-1	11.33 **	12.17 **	0.83	1.17	2.18 **	0.85	1.00	0.79		
L-6×T-1	0.01	−0.50	0.83	1.50	1.85 *	1.52	−2.56	−2.25		
L-7×T-1	−0.33	−3.00	0.01	−0.67	−0.32	−1.15	6.47 **	4.45 **		
L-8×T-1	−1.50	−10.00 **	0.17	1.83	−0.82	0.85	1.55	1.75		
L-9×T-1	−0.83	3.33	−1.34	−1.32	−0.65	−1.82 *	−1.98	−1.03		
L-10×T-1	−9.17 *	−5.83	−1.33	−1.33	1.35	1.35	−1.50	−1.88		
L-11×T-1	1.50	−0.83	0.50	−0.33	1.18	0.02	1.39	−3.75 *		
L-1×T-2	22.83 **	13.83 **	−2.67 *	−1.33	−1.68 *	−1.68 *	5.18 **	6.93 **		
L-2×T-2	−4.67	−8.50 *	1.00	1.83	1.48	2.48 **	1.88	1.03		
L-4×T-2	−17.17 **	−20.00 **	−0.33	0.33	0.15	0.82	−2.70	−0.03		
L-5×T-2	−11.33 **	−12.17 **	−0.83	−1.17	−2.18 **	−0.85	−1.00	−0.78		
L-6×T-2	0.01	0.50	−0.83	−1.50	−1.85 *	−1.52	2.57	2.25		
L-7×T-2	0.33	3.00	0.01	0.67	0.32	1.15	−6.47 **	−4.45 **		
L-8×T-2	1.50	10.00 *	−0.17	−1.83	0.82	−0.85	−1.55	−1.75		
L-9×T-2	0.83	−3.33	1.33	1.33	0.65	1.82 *	1.98	1.03		
L-10×T-2	9.17 *	5.83	1.33	1.33	−1.36	−1.34	1.50	1.88		
L-11×T-2	−1.50	0.83	−0.50	0.33	−1.18	−0.02	−1.38	3.75 *		

* and ** denote significance at the 0.05 and 0.01 levels of probability, respectively.

**Table 4 life-14-00453-t004:** Heterosis for number of rows/ear, number of grains/rows, and 100-kernel weight height relative to SC-10 and SC-30k8 under well-watered and stressed conditions.

Hybrid	Grain Yield/(t/ha)				Number of Rows/Ear			
	Relative to SC-10		Relative to SC-30k8		Relative to SC-10		Relative to SC-30k8	
	W-Watered	D-Stress	W-Watered	D-Stress	W-Watered	D-Stress	W-Watered	D-Stress
L-1×T-1	−19.83 **	−27.69 **	−21.17 **	−29.36 **	1.44	−11.25 *	2.72	−11.69 *
L-2×T-1	−5.08	−0.36	−6.67	−2.67	0.46	0.01	1.73	−0.50
L-4×T-1	−21.53 **	−16.21 **	−22.83 **	−18.15 **	−4.41	−2.50	−3.21	−2.99
L-5×T-1	0.17	7.10	−1.50	4.63	−19.53 **	−12.50 **	−18.52 **	−12.94 **
L-6×T-1	7.46	7.47	5.67	4.98	−2.46	−3.50	−1.23	−3.98
L-7×T-1	20.34 **	19.49 **	18.33 **	16.73 **	9.73	8.75	11.11	8.21
L-8×T-1	−3.73	2.55	−5.33	0.18	−11.73	−6.50	−10.62	−6.97
L-9×T-1	1.69	−7.10	0.09	−9.25	−7.34	−10.50 *	−6.17	−10.95
L-10×T-1	−17.80 **	−21.68 **	−19.17 **	−23.49 **	−7.34	−22.50 **	−6.17	−22.89 **
L-11×T-1	13.90 **	19.49 **	12.00 **	16.73 **	12.17	10.00 *	13.58 *	9.45 *
L-1×T-2	23.56 **	22.04 **	21.50 **	19.22 **	13.14	19.00 **	14.57 *	18.41 **
L-2×T-2	−10.17 *	−15.30 **	−11.67 **	−17.26 **	−0.02	5.00	1.23	4.48
L-4×T-2	−33.05 **	−28.96 **	−34.17 **	−30.60 **	4.85	20.00 **	6.17	19.40 **
L-5×T-2	−16.27 **	−17.49 **	−17.67 **	−19.40 **	7.78	2.50	9.14	1.99
L-6×T-2	−5.42	1.28	−7.00	−1.07	−1.49	5.00	−0.25	4.48
L-7×T-2	−10.34 *	−4.37	−11.83 **	−6.58	−0.02	−0.50	1.23	−1.00
L-8×T-2	−8.14 *	−18.58 **	−9.67 *	−20.46 **	0.95	1.50	2.22	1.00
L-9×T-2	−11.02 **	−13.30 **	−12.50 **	−15.30 **	−6.61	1.25	−5.43	0.75
L-10×T-2	−12.88 **	−18.03 **	−14.33 **	−19.93 **	−8.32	−10.50 *	−7.16	−10.95 *
L-11×T-2	12.71 **	20.77 **	10.83 **	17.97 **	4.85	22.50 **	6.17	21.89 **
**Hybrid**	**Number of grains/rows**				**100-kernel weight**			
	**Relative to SC-10**		**Relative to SC-30k8**		**Relative to SC-10**		**Relative to SC-30k8**	
	**W-Watered**	**D-Stress**	**W-Watered**	**D-Stress**	**W-Watered**	**D-Stress**	**W-Watered**	**D-Stress**
L-1×T-1	−5.19	−31.31 **	−6.10	−32.81 **	−10.99	−13.73 *	−15.24 *	−15.38 **
L-2×T-1	7.79	10.10	6.76	7.71	−11.99	−15.69 **	−16.19 *	−17.31 **
L-4×T-1	5.77	−5.05	4.76	−7.11	−17.99 *	−26.47 **	−21.90 **	−27.88 **
L-5×T-1	7.98	1.01	6.95	−1.19	−4.99	−14.71 **	−9.52	−16.35 **
L-6×T-1	6.73	8.28	5.71	5.93	−19.99 **	−29.41 **	−23.81 **	−30.77 **
L-7×T-1	18.46 **	5.05	17.33 **	2.77	12.01	9.80	6.67	7.69
L-8×T-1	4.62	3.03	3.62	0.79	4.01	−5.88	−0.95	−7.69
L-9×T-1	2.50	6.06	1.52	3.75	−4.99	−14.71 **	−9.52	−16.35 **
L-10×T-1	8.46	4.04	7.43	1.78	−0.99	−0.98	−5.71	−2.88
L-11×T-1	16.15 **	15.76 *	15.05 **	13.24 *	8.01	3.92	2.86	1.92
L-1×T-2	8.85	13.54 *	7.81	11.07	21.01	2.94	15.24 **	0.96
L-2×T-2	−32.69 **	−32.32 **	−33.33 **	−33.79 **	5.01	−10.78 *	0.01	−12.50 *
L-4×T-2	−61.54 **	−63.64 **	−61.90 **	−64.43 **	7.01	−19.61 **	1.90	−21.15 **
L-5×T-2	−18.46 **	−16.16 **	−19.24 **	−17.98 **	7.01	−10.78 *	1.90	−12.50 *
L-6×T-2	−2.12	1.01	−3.05	−1.19	8.01	−14.71 **	2.86	−16.35 **
L-7×T-2	−17.31 **	−16.36 *	−18.10 **	−18.18 **	1.01	−8.82	−3.81	−10.58
L-8×T-2	−5.38	−1.41	−6.29	−3.56	1.01	−6.86	−3.81	−8.65
L-9×T-2	0.77	4.75	−0.19	2.47	−7.99	−11.76 *	−12.38	−13.46 **
L-10×T-2	12.88	14.34 *	11.81 *	11.86	−1.99	−3.92	−6.67	−5.77
L-11×T-2	20.00 **	25.25 **	18.86 **	22.53 **	18.00	12.35 *	12.37	10.19

* and ** denote significance at the 0.05 and 0.01 levels of probability, respectively.

**Table 5 life-14-00453-t005:** Heterosis for plant height, ear height, days to tasseling, and days to silking relative to SC-10 and SC-30k8 under well-watered and stressed conditions.

Hybrid	Plant Height (cm)				Ear Height (cm)			
	Relative to SC-10		Relative to SC-30k8		Relative to SC-10		Relative to SC-30k8	
	W-Watered	D-Stress	W-Watered	D-Stress	W-Watered	D-Stress	W-Watered	D-Stress
L-1×T-1	−20.97 **	−19.69 **	−8.44 *	−9.20 **	−36.84 **	−33.84 **	−23.48 **	−23.53 **
L-2×T-1	−8.95 **	−13.65 **	5.48	−2.37	−24.40	−20.1	−8.41	−7.65
L-4×T-1	−2.69	−1.18	12.74 **	11.72 **	−7.66 *	−3.56	11.88 *	11.47 *
L-5×T-1	−1.79	−0.26	13.78	12.76	−12.20 **	−7.38	6.38	7.06
L-6×T-1	−4.09	−2.89	11.11 **	9.79 **	−3.11	1.27	17.39 **	17.06 **
L-7×T-1	−3.07	−2.23	12.30	10.53 **	−7.89 *	−7.12	11.59 *	7.35
L-8×T-1	−1.53	−0.92	14.07 **	12.02 **	−6.22	−13.49 **	13.62 **	0.01
L-9×T-1	−3.32	−3.02	12.00 **	9.64 **	−7.66 *	−3.31	11.88 *	11.76 *
L-10×T-1	−11.25 **	−10.10 **	2.81	1.63	−19.62 **	−17.56 **	−2.61	−4.71
L-11×T-1	−4.86	−4.86	10.22 **	7.57 **	−8.61 *	−6.87	10.72	7.65
L-1×T-2	0.51	−1.31	16.44 **	11.57 **	2.39	−5.85	24.06	8.82
L-2×T-2	−4.73	−4.86	10.37 **	7.57 **	−24.64 **	−26.21 **	−8.70	−14.71 **
L-4×T-2	−25.06 **	−23.75 **	−3.19 **	−13.80 **	−25.84 **	−27.23 **	−10.14 *	−15.88 **
L-5×T-2	−11.13 **	−10.76 **	2.96	0.89	−22.01 **	−19.08 **	−5.51	−6.47
L-6×T-2	9.08 **	10.24 **	26.37 **	24.63 **	3.35	8.91 *	25.22 **	25.88 **
L-7×T-2	−1.53	−0.26	14.07 **	12.76 **	−0.96	4.33	20.00 **	20.59 **
L-8×T-2	−2.56	−2.36	12.89 **	10.39 **	2.39	8.65 *	24.06 **	25.59 **
L-9×T-2	8.18 **	8.92 **	25.33 **	23.15 **	0.01	−1.53	21.16 **	13.82 **
L-10×T-2	−3.58	−3.15	11.70 **	9.50 **	0.01	−1.78	21.16 **	13.53 **
L-11×T-2	1.02	2.62	17.04 **	16.02 **	−4.31	1.27	15.94 **	17.06 **
**Hybrid**	**Days to tasseling**				**Days to silking**			
	**Relative to SC-10**		**Relative to SC-30k8**		**Relative to SC-10**		**Relative to SC-30k8**	
	**W-Watered**	**D-Stress**	**W-Watered**	**D-Stress**	**W-Watered**	**D-Stress**	**W-Watered**	**D-Stress**
L-1×T-1	2.22	3.98	3.95	8.28 *	−2.05	−1.57	−0.52	1.62
L-2×T-1	1.11	−3.98	2.82	0.01	−2.56	−4.71 *	−1.04	−1.62
L-4×T-1	−6.11 *	−4.55	−4.52	−0.59	−4.10	−4.71 *	−2.60	−1.62
L-5×T-1	−1.11	−1.70	0.56	2.37	−2.05	−1.57	−0.52	1.62
L-6×T-1	−6.11 *	−6.82 *	−4.52	−2.96	−4.62 *	−6.28 **	−3.13	−3.24
L-7×T-1	−5.56 *	−4.55	−3.95	−0.59	−7.18 **	−6.28 **	−5.73 **	−3.24
L-8×T-1	4.44	−6.25 *	6.21 *	−2.37	−3.08	−5.76 **	−1.56	−2.70
L-9×T-1	−1.67	−5.11	0.01	−1.18	−1.54	−3.14	0.01	0.01
L-10×T-1	−13.89 **	−13.64 **	−12.43 **	−10.06 **	−5.13 **	−8.38 **	−3.65	−5.41 *
L-11×T-1	−4.44	−3.41	−2.82	0.59	−5.13 **	−4.71 *	−3.65	−1.62
L-1×T-2	0.56	−2.27	2.26	1.78	−2.05	−1.57	−0.52	1.62
L-2×T-2	10.00 **	2.27	11.86 **	6.51 *	10.26 **	5.24 **	11.98 **	8.65 **
L-4×T-2	−2.22	−2.84	−0.56	1.18	3.59	1.05	5.21 **	4.32 *
L-5×T-2	−2.22	−1.70	−0.56	2.37	0.51	−3.14	2.08	0.01
L-6×T-2	−8.33 **	−6.82 *	−6.78 *	−2.96	−4.10 *	−6.81 **	−2.60	−3.78
L-7×T-2	−0.56	−1.70	1.13	2.37	1.54	0.01	3.13	3.24
L-8×T-2	1.11	−3.98	2.82	0.01	−0.51	2.09	1.04	5.41 *
L-9×T-2	5.56 *	2.27	7.34 **	6.51 *	9.23 **	4.19 *	10.94 **	7.57 **
L-10×T-2	−6.67 *	−6.25 *	−5.08	−2.37	−4.10 *	−7.33 **	−2.60	−4.32 *
L-11×T-2	−0.56	−2.27	1.13	1.78	0.01	−3.14	1.56	0.01

* and ** denote significance at the 0.05 and 0.01 levels of probability, respectively.

**Table 6 life-14-00453-t006:** Heterosis for chlorophyll content t relative to SC-10 and SC-30k8 under well-watered and stressed conditions.

Hybrid	Chlorophyll Content			
	Relative to SC-10		Relative to SC-30k8	
	W-Watered	D-Stress	W-Watered	D-Stress
L-1×T-1	0.95	−8.69	−1.84	−12.98 *
L-2×T-1	−9.34 *	−2.50	−11.84 **	−7.09
L-4×T-1	−2.23	−6.85	−4.93	−11.23 *
L-5×T-1	−4.40	−4.27	−7.04	−8.77
L-6×T-1	−1.96	2.65	−4.67	−2.18
L-7×T-1	16.04 **	11.93 *	12.83 **	6.67
L-8×T-1	−2.64	3.76	−5.33	−1.12
L-9×T-1	−14.07 **	−7.95	−16.45 **	−12.28 *
L-10×T-1	−13.19 **	−19.00 **	−15.59 **	−22.81 **
L-11×T-1	4.19	9.57	1.32	4.42
L-1×T-2	25.17 **	25.41 **	21.71 **	19.51 **
L-2×T-2	1.49	5.52	−1.32	0.56
L-4×T-2	−10.01 *	−3.53	−12.50 **	−8.07
L-5×T-2	−5.28	−4.27	−7.89	−8.77
L-6×T-2	11.64 *	16.05 **	8.55	10.60 *
L-7×T-2	−7.04	−4.27	−9.61 *	−8.77
L-8×T-2	−5.75	−0.52	−8.36	−5.19
L-9×T-2	−2.84	0.07	−5.53	−4.63
L-10×T-2	−3.92	−7.22	−6.58	−11.58 *
L-11×T-2	1.76	29.60 **	−1.05	23.51 **

* and ** denote significance at the 0.05 and 0.01 levels of probability, respectively.

**Table 7 life-14-00453-t007:** Stress tolerance indices for 22 maize hybrids assessed under well-watered and drought conditions.

Hybrid	Yp	Ys	MP	TOL	HM	SSI	STI	YI	YSI	GMP	YRR
L-1×T-1	9.0	7.6	8.30	1.40	8.24	2.37	0.59	0.76	0.84	8.27	15.56
L-2×T-1	10.7	10.4	10.55	0.30	10.55	0.43	0.96	1.03	0.97	10.55	2.80
L-4×T-1	8.8	8.8	8.80	0.00	8.80	0.00	0.67	0.88	1.00	8.80	0.00
L-5×T-1	11.3	11.2	11.25	0.10	11.25	0.14	1.09	1.11	0.99	11.25	0.88
L-6×T-1	12.1	11.2	11.65	0.90	11.63	1.14	1.17	1.11	0.93	11.64	7.44
L-7×T-1	13.5	12.5	13.00	1.00	12.98	1.13	1.46	1.24	0.93	12.99	7.41
L-8×T-1	10.8	10.7	10.75	0.10	10.75	0.14	1.00	1.06	0.99	10.75	0.93
L-9×T-1	11.4	9.7	10.55	1.70	10.48	2.28	0.96	0.97	0.85	10.52	14.91
L-10×T-1	9.2	8.2	8.70	1.00	8.67	1.66	0.65	0.82	0.89	8.69	10.87
L-11×T-1	12.8	12.5	12.65	0.30	12.65	0.36	1.38	1.24	0.98	12.65	2.34
L-1×T-2	13.9	12.8	13.35	1.10	13.33	1.21	1.54	1.27	0.92	13.34	7.91
L-2×T-2	10.1	8.9	9.50	1.20	9.46	1.81	0.78	0.89	0.88	9.48	11.88
L-4×T-2	7.5	7.4	7.45	0.10	7.45	0.20	0.48	0.74	0.99	7.45	1.33
L-5×T-2	9.4	8.6	9.00	0.80	8.98	1.30	0.70	0.86	0.91	8.99	8.51
L-6×T-2	10.6	10.6	10.60	0.00	10.60	0.00	0.97	1.05	1.00	10.60	0.00
L-7×T-2	10.1	10.0	10.05	0.10	10.05	0.15	0.87	1.00	0.99	10.05	0.99
L-8×T-2	10.3	8.5	9.40	1.80	9.31	2.67	0.76	0.85	0.83	9.36	17.48
L-9×T-2	10.0	9.1	9.55	0.90	9.53	1.37	0.79	0.91	0.91	9.54	9.00
L-10×T-2	9.8	8.6	9.20	1.20	9.16	1.87	0.73	0.86	0.88	9.18	12.24
L-11×T-2	12.7	12.6	12.65	0.10	12.65	0.12	1.38	1.25	0.99	12.65	0.79
S.C.10	11.2	10.5	10.85	0.70	10.84	0.95	1.02	1.04	0.94	10.84	6.25
S.C. 30K8	11.4	10.7	11.05	0.70	11.04	0.94	1.05	1.06	0.94	11.04	6.14

Yp: grain yield under well-watered condition (t/ha), Ys: grain yield under stressed condition (t/ha), MP: mean productivity, TOL: tolerance index, HM: harmonic mean, SSI: stress sensitivity index, STI: stress tolerance index, YI: yield index, YSI: yield stability index, GMP: geometric mean productivity, and YRR: yield reduction ratio.

## Data Availability

The data presented in this study are available upon request from the corresponding author.

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
