# Peer review of "Genetic Potential of Newly Developed Maize Hybrids under Different Water-Availability Conditions in an Arid Environment"

_life, 2024, doi:10.3390/life14040453_

Round 1
Reviewer 1 Report
Comments and Suggestions for Authors
All the comments are in the uploaded file.

Comments on the Quality of English LanguageComments are in the uploaded file.
Author Response
Dear Editor,
We would like to thank you and the reviewers for the time and efforts devoted to our manuscript entitled "Genetic Potential of Newly Developed Maize Hybrids in Agronomic Performance under Water Deficit and Well-Watered Conditions in Arid Environment" (ID: life-2886997). We have revised the manuscript according to the comments and suggestions the Reviewers pointed out. We have addressed comments point-by-point below in red. In addition, we have highlighted all the associated changes made to the manuscript using track changes.
Yours sincerely,
Authors
Responses to Reviewers' Comments
Reviewer 1:
The study conducted by Sedhom et al. explored the genetic potential of maize hybrids under water deficit conditions in arid environments. Drought and climate changes have dramatic effects on maize production in recent decades. The objective of the study was to evaluate the agronomic performance of maize hybrids – developed from selected inbred lines – in different environmental scenarios regarding water availability, and to highlight the potential tolerance to drought by selecting drought tolerant germplasm for future breeding programs and eventually for the maize production in hot, arid environments. Based on the agronomic performance in field trials the authors selected several inbred lines as a potential source of drought tolerant germplasm. Selected germplasm could be potential source for drought breeding programs.
Re: We would like to thank the Reviewer for his time dedicated to our manuscript and his positive assessment of our work.
Broad comments
The results from only one year and one location were used in the study, which narrows down
the probability to detect the tolerant germplasm. At the same time, that is the major constraint
of this study.
Re: This study was carried out during two seasons, in the first season the parental genotypes were cultivated and hybridized at the flowering stage using a line× tester mating design. During the second season the developed twenty top crosses along with their parental lines (ten lines and two testers) and two high-yielding commercial check hybrids, underwent initial evaluation under two distinct irrigation regimes. This step is essential for developing crosses and initial exploration to identify promising ones. The identified promising crosses will be evaluated through a multi-location, multi-year approach to thoroughly assess their stability and agronomic performance across a wider range of environmental conditions. This enhances the robustness of detecting tolerant germplasm across diverse scenarios to contribute valuable insights to maize cultivation.
The objective was to explore drought tolerance of maize hybrids in arid environment. If maize production is under full irrigation in such environments, drought tolerance is not so relevant.
Re: Egypt has a hot and arid climate with no precipitation events during the maize growing season. This arid climate requires drought-tolerant genotypes to be adapted to these predominant conditions, particularly under current climate change. However, the developed crosses were evaluated under well-watered and drought treatments to explore the promising ones under both conditions.
Egypt is a big maize producer. Are all maize growing regions under full irrigation? If so, maybe it is good to mention that as a common practice, and then comment about water scarcity in some regions (if you have some info comment about what is the percentage of maize regions with drought problems in Egypt).
Re: More information has been added as suggested (lines 116-119)
Consider the title change, e. g. 'Genetic potential of newly developed maize hybrids under water deficit and well-watered conditions in arid environment' or 'Genetic potential of maize hybrids under different water-availability conditions in arid environment' etc.
Re: The title has been modified as suggested
English is very good, it needs little attention (minor spelling check).
Re: Thanks for the positive assessment, the language has been revised throughout the manuscript.
Specific comments
Abstract
The objective is not stated in the abstract.
Re: The objective has been added as requested (lines 16-19)
Introduction
Line 53: low-yield heritability > 'low' is enough
Re: Done (line 71)
Materials and Methods
- 1. Parental genotypes and hybridization
Can you comment some more on the germplasm pedigree (i.e. kernel type, tropical/nontropical, heterotic patterns etc.)…?
Re: More details have been added as suggested (lines 100-103)
- 2. Experimental design and agronomic practices
Are two commercial checks known for drought tolerance? They should be mentioned in 2.1.
part and in Table 1 as well.
Re: The used checks are high-yielding hybrids (SC-10 and SC-30k8) adapted and widely cultivated across various Egyptian environments (lines 113-114).
What was plot length? It says in the manuscript that the width is 3x70 cm with 25 cm within
row. Plot length and plot size are needed.
Re: More details have been added as requested (lines 137-138)
What was the sample size for calculating grain yield? Formula for GY?
Re: More details have been added (lines 153-154)
Line 109: it says 3x70 cm and 0.25 m > put cm in both places
Re: The units have been unified as suggested
Results
Grain yield is the most important trait in maize breeding > it should be presented first in the
results, both in the text and the tables. Suggestion for the tables: put grain yield and yield-related traits first, then days to maturing, chlorophyll content etc.
- g. order for Tables 2, 3, 4, 5, 6, 7 and 8.
Grain yield -
Number of rows/ear
Number of grains/row
100 kernel weight
Plant height
Ear height
Days to tasseling
Days to silking
Chlorophyll content
It should be corrected in the text as well.
Re: The suggested order has been followed throughout the Tables, Figures, and text
It is maybe better to put 3.2. part at the end of the results.
Re: The section has been presented at the end as requested
Thanks so much for your review which contributed considerably to improve our manuscript.
Reviewer 2 Report
Comments and Suggestions for Authors
Dear Authors
Authors crossed Ten newly developed maize inbred lines with two high-yielding testers to assess the drought tolerance in the obtained 20 hybrids. The work provides interesting results especially introducing new lines that can be used in maize breeding programs. However, there are some errors need to be fixed.
>Using the shortened words such as GCA in the abstract is not recommended. Then until line 128 and 129 it is impossible to get what GCA and SCA stands for. In line 150-151 again I can see the full words and GCA and SCA. Please check all abbreviations in the manuscript.
>In table 1 there is no sufficient info about the lines if there is reference paper please add in the table or the GenBank passport data and information.
> There is not enough/clear info about the biological and experimental replicates and also outlier identification.
>In the results line 165 Equally, 165; “this top cross L-11×T-2 was the best for number of grains per row and 100 kernels weight under well-watered and stressed conditions (Table 4). The highest grain yield was produced by the cross L-1×T-2 under well-watered and water stress conditions followed by the cross L-11×T-2.”
If the cross L-11xT-2 had the best number of grains per row (It is not clear what authors means by telling the best, but I supposed it should be the highest number of grain) and also best 100 kernel weight (again probably should be the highest) then that means this hybrid should have the highest yield. Could you please explain then how L-1xT2 had the highest grain yield? These results that you provide here is a bit different than the discussion part in line 318 to 320. It might be due to writing error. To avoid that please first in result part give the top performed hybrids in control and then stress.
>Figure 4 is too much messy and complicated with not enough explanation in the legend. It will be better to provide PCs in different way.
>From line 139 to 144, it mentioned that some very important indices were calculated. However, I couldn’t find them anywhere. Instead of providing raw data in table 3, 4 and 5, I would prefer to see the main results for tolerance indices and those 3 tables with raw data can be provided as supplementary table.
>Line 290; The Principal
>Line 191; first two principal 290 components (PCAs) or PCs????? And also, again problem with abbreviations. You should give PCA in the M&M.
>Line 318; what do you mean by “In addition, the cross L-1×T-2 had the best value under well-watered conditions and cross L-11×T-2 under water stress for number of rows per ear.” What does best value means????
>Line 328; I would NOT talk about drought again. It is not introduction. In general discussion part is very weak. Needs to be written in better way and to be compared with other similar research in maize and/or other cereals.
Comments on the Quality of English Language
English language is good some minor corrections are needed.
Author Response
Dear Editor,
We would like to thank you and the reviewers for the time and efforts devoted to our manuscript entitled "Genetic Potential of Newly Developed Maize Hybrids in Agronomic Performance under Water Deficit and Well-Watered Conditions in Arid Environment" (ID: life-2886997). We have revised the manuscript according to the comments and suggestions the Reviewers pointed out. We have addressed comments point-by-point below in red. In addition, we have highlighted all the associated changes made to the manuscript using track changes.
Yours sincerely,
Authors
Responses to Reviewers' Comments
Reviewer 2:
Dear Authors
Authors crossed Ten newly developed maize inbred lines with two high-yielding testers to assess the drought tolerance in the obtained 20 hybrids. The work provides interesting results especially introducing new lines that can be used in maize breeding programs. However, there are some errors need to be fixed.
Re: We would like to thank the Reviewer for the time dedicated to our manuscript and positive assessment of our work.
Using the shortened words such as GCA in the abstract is not recommended. Then until line 128 and 129 it is impossible to get what GCA and SCA stands for. In line 150-151 again I can see the full words and GCA and SCA. Please check all abbreviations in the manuscript.
Re: Thank you for your feedback. Based on your suggestion, We have ensured that all abbreviations are fully spelled out at their first mention for clarity.
In table 1 there is no sufficient info about the lines if there is reference paper please add in the table or the GenBank passport data and information.
Re: More details have been added to Table 1
There is not enough/clear info about the biological and experimental replicates and also outlier identification.
Re: More details have been added (line 134)
In the results line 165 Equally, 165; "this top cross L-11×T-2 was the best for number of grains per row and 100 kernels weight under well-watered and stressed conditions (Table 4). The highest grain yield was produced by the cross L-1×T-2 under well-watered and water stress conditions followed by the cross L-11×T-2." If the cross L-11xT-2 had the best number of grains per row (It is not clear what authors means by telling the best, but I supposed it should be the highest number of grain) and also best 100 kernel weight (again probably should be the highest) then that means this hybrid should have the highest yield. Could you please explain then how L-1xT2 had the highest grain yield? These results that you provide here is a bit different than the discussion part in line 318 to 320. It might be due to writing error. To avoid that please first in result part give the top performed hybrids in control and then stress.
Re: The crosses L-1×T-2, L-11×T-2, and L-7×T-1 displayed the highest grain yield under well-watered and water stress conditions. Also, these top crosses were the best among all studied hybrids for number of rows per ear, number of grains per row and 100 kernels weight under well-watered and stressed conditions. The paragraph has been revised and improved (lines 185-189) as well as the corresponding discussion has been revised (lines 431-436)
Figure 4 is too much messy and complicated with not enough explanation in the legend. It will be better to provide PCs in different way.
Re: More explanations have been added to the legend (lines 412-415)
From line 139 to 144, it mentioned that some very important indices were calculated. However, I couldn't find them anywhere. Instead of providing raw data in table 3, 4 and 5, I would prefer to see the main results for tolerance indices and those 3 tables with raw data can be provided as supplementary table.
Re: The tolerance indices have been added in Table 7 (line 385) as requested
Line 290; The Principal
Re: "principle" has been replaced by "principal" (line 395)
Line 191; first two principal components (PCAs) or PCs????? And also, again problem with abbreviations. You should give PCA in the M&M.
Re: The abbreviations have been revised (lines 397, 401), and all abbreviations throughout the manuscript have been addressed
Line 318; what do you mean by "In addition, the cross L-1×T-2 had the best value under well-watered conditions and cross L-11×T-2 under water stress for number of rows per ear." What does best value means????
Re: The sentence has been revised and improved to be “Specific crosses displayed promising agronomic performance under both well-watered and drought conditions. Remarkably, the crosses L-1×T-2, L-11×T-2, L-7×T-1, and L-11×T-1 exhibited superior grain yield, number of rows per ear number of grains per row, and 100-kernel weight under both conditions. These crosses represent promising candidates for developing drought-resilient maize hybrids” (lines 431-436)
Line 328; I would NOT talk about drought again. It is not introduction. In general discussion part is very weak. Needs to be written in better way and to be compared with other similar research in maize and/or other cereals.
Re: The discussion has been revised and improved as requested
English language is good some minor corrections are needed.
Re: The manuscript has been carefully revised, and the language has been improved
Thanks so much for your review which contributed considerably to improve our manuscript.
Reviewer 3 Report
Comments and Suggestions for Authors
It is intetesting evaluation of drought resistant traits in maize. The text well-written. Some minor corrections are required.
Line 11: Here should be words “in the current paper we described ten newly…
Summary: there are too much lines name as numbers. Summary must reflect more general finding, not too much details.
Line 68: “ Several breeders detected favorable heterotic effects for maize grain yield and contributing agronomic traits [14-19].” – I think it is already well-know fact long long ago..
Lines 97: pH 7,9 is not a slightly alakaline, it is rather middle alkaline.
Line 147- 150_ please, provide some introduction to this part. Do not start with numbers/table!
Line 200: “highest negative” ¿?
Line 238: “negative and significant heterotic” ¿??
Line 276: “the second and third best heterotic effect” ¿??
Comments on the Quality of English Languageminor editing
Author Response
Dear Editor,
We would like to thank you and the reviewers for the time and efforts devoted to our manuscript entitled "Genetic Potential of Newly Developed Maize Hybrids in Agronomic Performance under Water Deficit and Well-Watered Conditions in Arid Environment" (ID: life-2886997). We have revised the manuscript according to the comments and suggestions the Reviewers pointed out. We have addressed comments point-by-point below in red. In addition, we have highlighted all the associated changes made to the manuscript using track changes.
Yours sincerely,
Authors
Responses to Reviewers' Comments
Reviewer 3:
It is intetesting evaluation of drought resistant traits in maize. The text well-written. Some minor corrections are required.
Re: We would like to thank the Reviewer for the time dedicated to our manuscript and positive assessment of our work.
Line 11: Here should be words "in the current paper we described ten newly…
Re: More details have been added (lines 16-19)
Summary: there are too much lines name as numbers. Summary must reflect more general finding, not too much details.
Re: The abstract has been revised and improved
Line 68: "Several breeders detected favorable heterotic effects for maize grain yield and contributing agronomic traits [14-19]." – I think it is already well-know fact long long ago.
Re: The sentence has been removed (lines 87-88)
Lines 97: pH 7,9 is not a slightly alakaline, it is rather middle alkaline.
Re: Modified as suggested (line 122)
Line 147- 150_ please, provide some introduction to this part. Do not start with numbers/table!
Re: Done as requested (lines 177-179)
Line 200: "highest negative"?
Re: Modified to "significant negative GCA effects" (line 244)
Line 238: "negative and significant heterotic"??
Re: Modified to "significant negative heterosis" (line 320)
Line 276: "the second and third best heterotic effect"??
Re: modified to "The cross L-7×T-1 exhibited the highest heterosis for grain yield relative to SC-10 and SC-30k8 under both environments followed by L-11×T-2” (lines 301-303)
Comments on the Quality of English Language: minor editing:
Re: The manuscript has been carefully revised, and the language has been improved
Thanks so much for your review, which contributed considerably to improve our manuscript.
Round 2
Reviewer 1 Report
Comments and Suggestions for Authors
The paper is now improved, however, there is still issue of one year of testing on one location.
Author Response
Dear Editor,
We extend our sincere gratitude to you and the Reviewers for dedicating time and effort to our manuscript titled "Genetic Potential of Newly Developed Maize Hybrids in Agronomic Performance under Water Deficit and Well-Watered Conditions in Arid Environment" (ID: life-2886997). We sincerely appreciate the review process which has enhanced the clarity and overall quality of our manuscript.
We have addressed the new comments provided by the reviewers below.
Thank you once again for your invaluable support and guidance.
Yours sincerely,
Authors
Responses to Reviewers' Comments
Reviewer 1:
The paper is now improved, however, there is still issue of one year of testing on one location.
Re: We highly appreciate dedicating time to review our manuscript and providing positive feedback on our previous revisions.
Our plant materials in this study were newly developed inbred lines derived from ten generations of self-pollination. These lines were cultivated and hybridized during the first season of this study. Subsequently, in the second season, twenty crosses, along with their twelve parental lines and two high-yielding commercial check hybrids, were subjected to evaluation under two distinct water availability conditions in dry summer climate conditions. This is a crucial step in identifying promising crosses that lay the foundation for further exploration. The promising crosses identified will be evaluated through a multi-location, multi-year approach. These processes will provide tolerant maize germplasm suitable for maize cultivation under arid conditions.
The revised version has more clarification (lines 99-117).
Once again, we thank the Reviewer for his valuable feedback and support.
Reviewer 2 Report
Comments and Suggestions for Authors
Dear Authors
Thank you for addressing my comments. If you have the filed capacity values for well watered and stressed conditions it would be great to add them too. Line 127, First the sentences "While drought treatment was applied every 21 days" is grammatically incorrect and why and how did you considered irrigation every 21 days as a stress condition?
Best wishes
Author Response
Dear Editor,
We extend our sincere gratitude to you and the Reviewers for dedicating time and effort to our manuscript. We sincerely appreciate the review process which has enhanced the clarity and overall quality of our manuscript.
We have addressed the new comments provided by the reviewers below in red.
Thank you once again for your invaluable support and guidance.
Yours sincerely,
Authors
Responses to Reviewers' Comments
Reviewer 2:
Thank you for addressing my comments. If you have the filed capacity values for well watered and stressed conditions it would be great to add them too. Line 127, First the sentences "While drought treatment was applied every 21 days" is grammatically incorrect and why and how did you considered irrigation every 21 days as a stress condition?.
Re: We appreciate the Reviewer for dedicating time to assess our manuscript and providing positive feedback on our previous revisions.
Regarding the concern about stress, we followed the recommended irrigation practices for maize in the study region as determined by the Department of Water Requirement and Field Irrigation, Center of Agricultural Research under the Egyptian Ministry of Agriculture and Land Reclamation. The optimal irrigation amount was identified at approximately 7900 m3/ha based on climatic variables and soil type. Accordingly, we applied irrigation to the well-watered treatment every 12 days providing a total of 7990 m3/ha throughout the growing season. Conversely, irrigation in the other treatment was applied every 21 days, supplying 5411 m3/ha, which represents about 65% of the recommended amount to induce water deficit conditions.
The revised version has more clarification (lines 128-136).
Once again, we thank the Reviewer for his valuable feedback and support.